# *In vitro* culture of human-infecting *Encephalitozoon* spp. for genome sequencing with minimal host contaminant

**Anne Caroline Mascarenhas dos Santos**, **Pingdong Liang**, **Oscar X. Juárez**, **Jean-François Pombert**, **Karina Tuz***

Biology Department, Illinois Institute of Technology, Chicago, Illinois, United States of America

* ktuz@illinoistech.edu

## Abstract

Microsporidia are obligate intracellular parasites infecting a wide range of hosts, including humans. They can cause severe infectious diseases if left untreated, particularly in immunocompromised individuals. The propagation of the human-infecting microsporidian species *in vitro* is essential for generating sufficient material for genomics studies, yet existing protocols often lack detail, accessibility, or strategies to minimize host DNA contamination. Here, we present an optimized, reproducible protocol for culturing *Encephalitozoon* spp. in human foreskin fibroblasts (HFF-1), designed to produce high-yield and high-quality genomic DNA with minimal host DNA for downstream sequencing experiments. In one month, our method yielded approximately one billion spores, which were purified using mechanical disruption, filtration, detergent treatment, and DNase I treatment to remove free host DNA. The reduction of host DNA was validated through PCR, and next-generation sequencing – with Illumina, PacBio and Oxford Nanopore – revealing that 83–97% of the reads mapped to microsporidian genomes. This protocol enabled the generation of the first telomere-to-telomere assemblies for *E. cuniculi*, *E. hellem,* and *E. intestinalis*. Our workflow provides a robust framework for producing microsporidian genomic material suitable for advanced genomics applications, with potential adaptability to other intracellular pathogens.

## Introduction

Microsporidia are a diverse group of spore-forming obligate intracellular parasites well-known to infect a wide range of hosts, from insects to mammals [1–3]. These organisms are highly problematic in the apiculture, sericulture, and aquaculture industries wherein widespread infections can lead to serious economic harm [4–6]. Microsporidia are also opportunistic human pathogens with several species known to affect both immunocompromised and, to a lower extent, immunocompetent

**Data availability statement:** All relevant data are within the manuscript and its Supporting Information files.

**Funding:** This work was supported by the National Institute of Allergy and Infectious Diseases of the National Institutes of Health [grant number R15AI128627] to J.-F.P. and [grant number 1R01AI151152-01A1] to O.J and K.T. The content is solely the responsibility of the authors and does not necessarily represent the official views of the National Institutes of Health. The founders had no role in study design, data collection and analysis, decision to publish, or preparation of the manuscript. There was no additional funding received for this study.

**Competing interests:** The authors have declared that no competing interests exist.

individuals. Infections in patients can be localized and/or disseminated; symptoms often include chronic diarrhea, bronchitis, conjunctivitis and/or encephalitis [7].

Given their obligate parasitic lifestyle, microsporidia cannot be cultivated outside of their hosts (at least for now), and studies requiring biological materials rely on either direct isolation from infected hosts or from *in vitro* propagation in cell cultures. Except for insect-related studies wherein live host growth is both ethically sound and feasible within a reasonable timeframe, for applications requiring large amounts of material, the propagation in cell cultures is usually the only available solution. Although cell culture propagation of human-infecting microsporidia has been reported in the literature for over two decades [8], the complete protocols are not always easily available. Some are shortened to fit to publication page limitations and thus often lack the minutiae of details required for a non-expert to reproduce while others are only accessible behind a paywall due to the journal or book publisher closed/toll access policies. Additionally, when accessible, the protocols are frequently rewritten in a format not directly usable for wet lab applications, with the reformatting often resulting in a loss of valuable information.

There are protocols available with enough detail to culture *Encephalitozoon* spp. in different cells lines [9,10], however, microsporidian spore yield nor the specific cultivation times are described, which is key information to offer guidance for enough starting material for genomic extraction for sequencing analyses. We encountered these issues while designing the protocols used for our microsporidia genome sequencing studies [8,9,11] and had to resort to several rounds of trial and error and standardization to determine the proper conditions for consistency and reproducibility. Additionally, none of the published protocols included steps to minimize host DNA contaminant for downstream genomics applications, which is critical to maximize sequencing outputs. To assist those looking to start working with microsporidia cell cultures and promote accessible, reproducible methods, we outline our protocols below, with lab-ready versions in the supplementary information. Using these protocols, we were able to obtain approximately 1 billion microsporidian spores in one month of culturing, which yielded high quality microsporidia genomic DNA (gDNA) with minimal host DNA contaminant and culminated in the release of the first telomere-to-telomere Microsporidia genomes [12].

## Materials and methods

The protocol described in this peer-reviewed article is published on protocols.io (https://doi.org/10.17504/protocols.io.j8nlk1mz5g5r/v1) and is included for printing purposes as S1 File.

### Human foreskin fibroblast cell culture infected with *Encephalitozoon* spp

Murine and human fibroblasts are readily infected with *Encephalitozoon* spp. [9,13–18], therefore, we established the cultivation protocol in HFF-1 cells. Spores of human-infecting *Encephalitozoon* spp. from the American Type Culture Collection (*Encephalitozoon intestinalis* ATCC 50506, *Encephalitozoon hellem* ATCC 50451,

*Encephalitozoon hellem* ATCC 50604, and *Encephalitozoon cuniculi* ATCC 50602) can be propagated *in vitro* in HFF-1 cells (ATCC SCRC-1041) at 37˚C and 5% $CO_2$ in a humidified atmosphere.

HFF-1 cells were cultured on 100 mm petri dishes (Thermo Scientific Cat# 150350) coated with 0.1% (w/v) gelatin from bovine skin (Sigma-Aldrich Cat# G9391) (protocol 2.1 supporting information) in Dulbecco's Modified Eagle Media (DMEM) (Gibco Cat# 11995−065) enriched with 10% (v/v) heat-inactivated fetal bovine serum (FBS) (Cytiva Cat# SH30070.03), 1% (v/v) PSQ 100x (10,000 units penicillin [final concentration of 100 units] + 10,000 µg streptomycin [final concentration of 100 µg] + 29.2 mg/mL L-glutamine [final concentration of 0.29 mg/mL] in 10 mM citrate buffer, Gibco Cat# 10378016) and 2 mM L-glutamine (Cytiva Cat# SH40003.01) (protocol 1 supporting information) (Table 1). The cells were fed every two days by half media changes to reduce stress and preserve secreted factors (protocol 2.3.1 supporting information).

Once the HFF-1 cell monolayer was confluent (90%−100%; approximately 1 week, Fig 1), the cell culture washed with phosphate-buffered saline (PBS 1X, pH 7.4) (Gibco Cat# 14190) to remove leftover media. Cells were detached with 3 mL of 0.05% trypsin/EDTA (Gibco Cat# 25300054) for 90 s at 37˚C and 5% CO2, then trypsin was inactivated with 3 mL of fresh enriched DMEM at room temperature (RT) for 2.5 min. Cells were pelleted by centrifugation at 1,000 x *g* for 5 min at RT and resuspended in 1 mL of fresh enriched DMEM by gentle pipetting. Cells were seeded at a density of 2.5 x $10^4$ cells/$cm^2$, a total of 500 µL of cell suspension was transferred to each two new gelatin-coated 10 mm petri dishes containing 10 mL of fresh media and incubated at 37˚C and 5% $CO_2$ in a humidified atmosphere (protocol 2.3.2 supporting information). Cell cultures were monitored daily for adherence, confluence and contamination under the inverted phase-contrast Eclipse Ts2 Nikon microscope (Minato City, Tokyo, Japan) (Fig 1).

After subpassage, HFF-1 cells were infected with *Encephalitozoon* spores from different species: *Encephalitozoon intestinalis* ATCC 50506, *Encephalitozoon hellem* ATCC 50451, *Encephalitozoon hellem* ATCC 50604, and *Encephalitozoon cuniculi* ATCC 50602. The same method was used for all species. HFF-1 cells were infected with microsporidia by quickly thawing the ATCC vials containing the *Encephalitozoon* spp. clinical samples in a 37˚C water bath (1−2 min) and immediately transferring the contents to the confluent cell culture monolayers by pipetting. The multiplicity of infection (MOI) range varied from 0.5 to 20. The microsporidia sample must be added to the cell culture media rather than the bottom of the petri dish, so that the spores can diffuse easily across the culture. Petri dishes were rocked to ensure an even spread of the spores and incubated at 37˚C with 5% $CO_2$ in a humidified atmosphere. The cell culture media (enriched DMEM) was entirely replaced 24 hours post-infection (hpi) to remove the dimethyl sulfoxide (DMSO) from the

**Table 1. Materials used for HFF-1 cell culture and microsporidian spore isolation.**

| Material | Manufacturer | Catalog number |
|---|---|---|
| DMEM<br> + 4.5 g/L D-Glucose<br> + L-Glutamine<br> + 110 mg/L Sodium pyruvate | Gibco | 11995−065 |
| FBS | Cytiva | SH30070.03 |
| L-Glutamine | Cytiva | SH40003.01 |
| PSQ | Gibco | 10378016 |
| DPBS (no $CaCl_2$, no $MgCl_2$) | Gibco | 14190 |
| 0.05% Trypsin/EDTA | Gibco | 25300054 |
| Gelatin from bovine skin Type B | Sigma-Aldrich | G9391 |
| Tween 20 | Fisher scientific | BP337 |
| DNase I from bovine pancreas | Sigma-Aldrich | DN25 |
| Petri dishes | Thermo Scientific Nunc | 150350 |

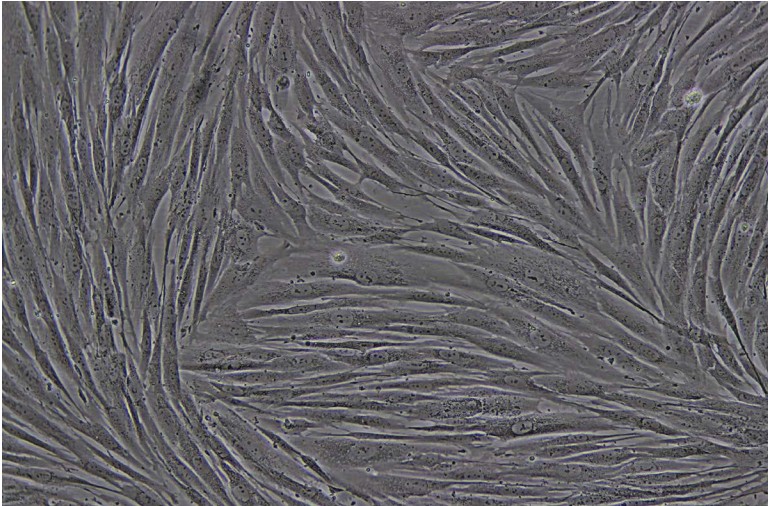

**Fig 1. Human foreskin fibroblast (HFF-1) cell culture monolayer.** HFF-1 cells 100% confluent observed under phase-contrast inverted microscope with a 40X magnification and the Canon EOS REBEL T3i digital camera.

microsporidia ATCC cryogenic media (protocol 2.4 supporting information). DMSO is a common cryoprotectant used to store cell lines that can increase cell death and thus should be removed whenever possible [19]. The culture media of microsporidia-infected cell line (Figs 2 and 3) should be changed when it gets cloudy due to high spore density or when it

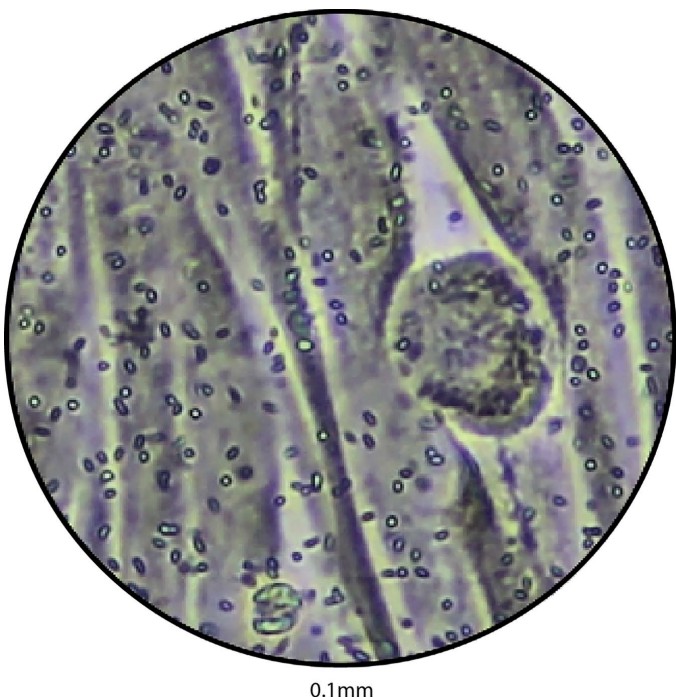

0.1mm

**Fig 2. *Encephalitozoon intestinalis* ATCC 50506 spores in HFF-1 cell culture.** Culture visualized under phase-contrast inverted microscope with 10X objective and Canon EOS REBEL T3i digital camera. Each spore is about 1 μm wide to 3.5 μm in length.

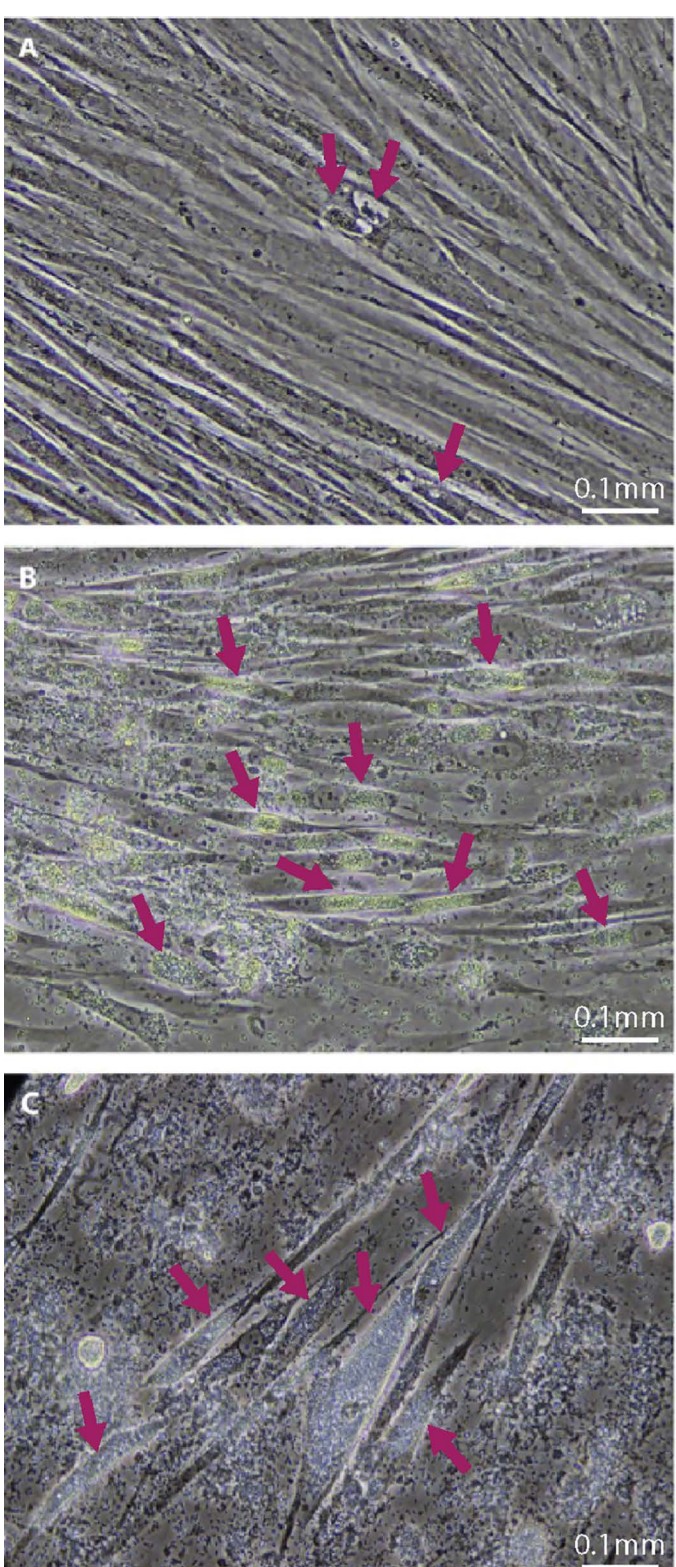

**Fig 3. HFF-1 cells infected with *Encephalitozoon hellem* ATCC 50451 at different post- infection days.** *E. hellem* vacuoles in HFF-1 cell culture (magenta arrow) at 7 dpi **(A)**, 27 dpi **(B)**, and 32 dpi (harvest stage) **(C)**.

changes from red to orange color due to acidification from metabolites (protocol 2.5.1 supporting information). Note that the microsporidian spore load can be affected if media is replaced too often, potentially decreasing the spore density in the cell culture. Microsporidian spores can be harvested from the cell culture supernatant by centrifugation (1,500 x $g$, 20 min, RT) and resuspended in 10 mL of PBS (1X) for purification or for storage at 4°C (protocol 2.5.2 supporting information). The microsporidia-infected HFF-1 cell cultures were checked daily for signs of infection and for the presence of microsporidian spores using an inverted phase-contrast microscope with magnifications of 400X and 800X (Fig 2).

The starter microsporidia-infected HFF-1 cell culture was split when the confluence of infected cells reached 90–100%. In the protocol that we optimized for these cultures, the culture supernatant is kept before culture splitting, together with the two 3 mL PBS (1X) washes, to harvest spores at a later stage instead of discarding it (thus increasingly the overall spore yield). Following the PBS washes, cells were detached with 3 mL of trypsin/EDTA (0.05%) followed by incubation at 37°C in the 5% $CO_2$ incubator for 90 sec –longer trypsinization was required for older cultures to ensure cells were detached from the plate – and visual inspection using the phase contrast microscope. Trypsin was inactivated by addition of 3 mL of fresh enriched DMEM and incubation at RT for 2.5 min. The cell suspension was pelleted by centrifugation (1,000 x $g$, 5 min, RT), and resuspended in 1 mL of fresh media, and 100 μL were transferred to 10 gelatin-coated petri dishes (100 mm) containing 10 mL of enriched DMEM each (protocol 2.5.3 supporting information).

The microsporidia cell culture was maintained until all 10 petri dishes reached 90–100% infection (approximately a month), following the herein described protocols. Fig 3 shows *E. hellem* ATCC 50451 infected HFF-1 cells (magenta arrows) after different times post-infection: at 7 days post-infection (dpi), we observed the formation of vacuoles in a few HFF-1 cells, as anticipated [8,9] (A); at 27 dpi most of the cells in the culture presented signs of infection by microsporidia (B); and at 32 dpi the spore-containing vacuoles showed a significant increase in size, occupying most of the host cell cytosol (C). As observed in Fig 3C, the majority of the cells were destroyed in about a month of culturing, a faster timeline than the reported for *Encephalitozoon* spp. propagation [8].

It is important to disclose that after a month of maintenance, the microsporidia infected HFF-1 cells were no longer organized in a monolayer. However, the goal of these protocols was to produce sufficient microsporidian spores, for the extraction of clean gDNA (500 million spores generated ~3 μg of HMW gDNA of at least 80% purity) to perform high-throughput sequencing, subsequently, the status of the HFF-1 cells (monolayer or not) was not problematic. No differences on spore production were observed among the different *Encephalitozoon* spp. used. Because high density cell layouts might be detrimental to other cell biology experiments (*e.g.,* immunostaining), the protocols for keeping HFF-1 cell cultures infected with microsporidia should be adjusted as needed.

## Isolation of microsporidian spores

Microsporidian spores were purified when 90−100% of HFF-1 cells showed signs of infection (described in section 2.2; protocol 3.1 supporting information). The cell culture supernatant containing spores was harvested for later DNA extractions (protocol 2.5.2 supporting information) and the HFF-1 cells from each petri dish were washed with 3 mL of PBS twice to remove leftover DMEM, PBS washes were collected and pooled together with the initial media supernatant. Infected HFF-1 cells were treated with 3 mL trypsin/EDTA (0.05%) and incubated at 37°C in a 5% $CO_2$ incubator for 90 sec or until cells were completely detached, trypsin was inactivated by the addition of a similar amount of fresh media.

Detached infected-HFF-1 cells were passed through a 27-gauge (27-G) needle using a 10 mL syringe thrice (up and down) to break the cells and release the microsporidian spores. The lysed cell suspension was passed through a 5 μm polyvinylidene difluoride (PVDF) filter to minimize host cells debris in the sample; *Encephalitozoon* spp. spores are 3.5 μm in length and easily passed through the filter.

Spores were pelleted by centrifugation (1,500 x $g$, 20 min, RT) and resuspended in 1 mL of PBS-Tween 20 (0.3%, filter sterilized) to help solubilize leftover HFF-1 cell membrane in the sample [20]. Spores were pelleted by centrifugation (1,500 x $g$, 20 min, RT) and washed with 1 mL of PBS to prevent Tween 20 (Fisher Scientific Cat# BP337) remnants from

affecting downstream procedures. Spores were resuspended in 1 mL of PBS, counted using a hemocytometer (sample aliquot was diluted to a 1:100 ratio; Fig 4). Spore samples were stored at 4°C (spores remained viable for up to 3 years). To verify the yield of purified microsporidian spores generated by this protocol, we counted the number of spores obtained from a single plate of 100% confluent HFF-1 cells infected with *E. intestinalis* ATCC 50506 using a hemocytometer (Fig 4). The results showed that after a month of culturing, we were able to obtain approximately 150 million spores from a single 100 mm cell culture petri dish.

Spores can also be purified from the cell culture supernatant (protocol 3.2 supporting information) using a protocol similar to the purification from infected host cells (steps 1–5, protocol 3.1, supporting information). Spores are collected from the liquid media during renewal step, which was followed by pelleting of spores through centrifugation (1,500 x *g*, 20 min, RT) and resuspension in 10 mL of PBS (1X) by pipetting. The cell breakage, host cell debris filtering, and host cell membrane solubilization steps were similar to the purification of spores from infected host cells. There was no need to extract spores from the cell culture supernatant, since the spores obtained from the infected host cells were more than sufficient for our purposes (1 billion spores from 10 petri-dishes). However, these spores could be used to generate stocks of purified microsporidia for future cell culture applications.

## DNase I treatment of microsporidian purified spores

Prior to DNA extraction, the stored and purified spores consisted of a "sticky" difficult-to-resuspend white-colored pellet, indicating the presence of significant free DNA in the spore suspension (pure microsporidian spore pellets are usually white-colored and easy to resuspend), likely from the host sample. To assess the sample composition, total HMW gDNA (1.14 µg) was extracted from *E. intestinalis* spores (150 million or 1 petri-dish) and resuspended

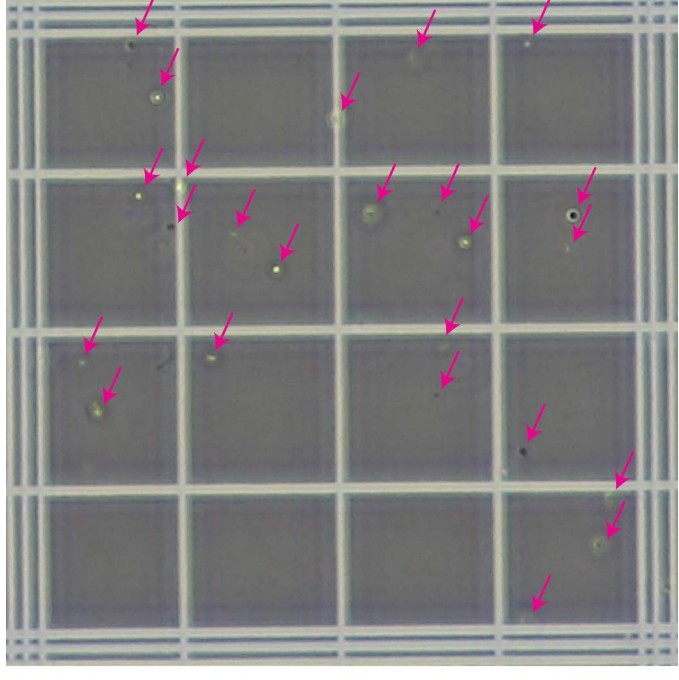

0.25mm

**Fig 4.** *Encephalitozoon cuniculi* **ATCC 50602 purified spores.** Single spores are marked by magenta arrows. Spore suspension was diluted to a 1:100 ratio and visualized/counted in hemocytometer under phase-contrast microscope with 10X objective and Canon EOS REBEL T3i digital camera.

in molecular biology grade (MBG) water using previously described protocols [12,21]. Briefly, pelleted spores were resuspended in 300 µl of lysis solution from the MasterPure Complete DNA Purification Kit (Biosearch Technologies, UK) containing proteinase K and mixed thoroughly. Glass beads (20 µl, 150–212 µm in diameter) were added to the samples, which were immediately incubated at 65°C for 15 min and bead beaten 30 s every 5 min to ensure the release of DNA from the chitin-walled spores. Finally, the supernatant was subjected to DNA purification following the manufacturer's guidelines for the MasterPure Complete DNA Purification Kit and resuspended in ultra-pure molecular biology grade water [12,21]. Then, we amplified bacteria, mammal, and microsporidian DNA by polymerase chain reaction (PCR) for a total of 40 cycles (initial denaturation: 95°C for 30 s; denaturation: 95°C for 30 s; annealing: 55°C for 30 s; elongation: 72°C for 2 min; final elongation: 72°C for 5 min). Universal 16S bacteria primers (357F-1100R), universal mammal ribosomal RNA (rRNA) primers (MF787-MR7535), and three sets of microsporidia rRNA primers (lsuF1- lsuR1, lsuF2-lsuR2, and ssuF1-ssuR1) specific for the small and large subunits (ssu and lsu) of both *E. intestinalis* and *E. hellem* (Table 2) were used for amplification [22]. The amplicons were verified by agarose gel electrophoresis (0.8%, TAE buffer 1X, 75 volts; Fig 5).

The PCR results showed that microsporidian DNA was present in the sample (wells 1–3, Fig 5) but that human DNA (well 5, Fig 5) was more abundant in the sample. These results confirmed that contaminant DNA from the host was present in the spore suspension and that it needed to be minimized prior to extraction of microsporidian genomic DNA for downstream sequencing analysis. To minimize the host DNA content in the sample, we performed a DNase I treatment (protocol 4 in supporting information) to degrade only free DNA in the sample. The gDNA from microsporidian spores was not affected because it was protected by the chitin walls of the unbroken spores.

To treat the sample with DNase I, the spore sample (spores purified from 10 x 100 mm petri dishes of infected host cells in 10 mL of PBS) was resuspended by vortexing, and 10 µL of DNase I (10 mg/mL) (Sigma-Aldrich Cat# DN25) together with 15 µL of $MgCl_2$ (1 M, filter sterilize) were added to the suspension (DNase I requires $MgCl_2$ as a cofactor to be functional). The sample was incubated at RT for 15 min under rotation with a tube revolver (hula-mixer) with a fixed speed of 22 rpm (Crystal Technology & Industries, Addison, TX, USA). DNase I activity was stopped by mixing 60 µL of EDTA (0.5 M, pH8, filter sterilized) with the spore sample through inversion (EDTA is a chelating agent and sequesters $MgCl_2$ from DNase I). After this step, free-DNA was degraded, and microsporidian spores were

**Table 2. Primers [22] used to amplify bacterial, mammal, and microsporidian DNA.**

| Primer | Sequence (5'→3') | Length (nt) | GC (%) | Tm (˚C) | Expected Amplicon Size (bp) |
|--------|------------------|-------------|--------|---------|------------------------------|
| 357F | CTCCTACGGGAGGCAGCAG | 19 | 68.4 | 60.0 | 750 |
| 1100R | AGGGTTGCGCTCGTTG | 16 | 62.5 | 56.1 | |
| MF787 | ATGCTCTTAGCTGAGTGTCC | 20 | 50.0 | 54.2 | 1,000 |
| MR7535 | GCACTTACTGGGAATTCCTC | 20 | 50.0 | 53.0 | |
| ss530F | GTGCCAGCMGCCGCGG | 16 | 84.4 | 65.6[a] | 500 |
| ss1047R | AACGGCCATGCACCAC | 16 | 62.5 | 56.6 | |
| lsuF1 | GGGAGTAACTATGACTCTC | 19 | 47.4 | 48.4 | 100 |
| lsuR1 | GGTTCACAATACARGTCG | 18 | 47.2 | 49.2[b] | |
| lsuF2 | GACAGAAAAGTTACCACAG | 19 | 42.1 | 48.1 | 100 |
| lsuR2 | TCACGATGAGAAGAG | 15 | 46.7 | 43.5 | |
| ssuF1 | KGTCCCTGTCCTTTGTAC | 18 | 52.8 | 51.8[c] | 100 |
| ssuR1 | CGACTTATATCTTATCTAACACGA | 24 | 33.3 | 49.6 | |

[a]Average Tm. The degenerated primer has a minimal Tm of 64.1˚C and maximum Tm of 67.2˚C.

[b]Average Tm. The degenerated primer has a minimal Tm of 48.1˚C and maximum Tm of 50.3˚C.

[c]Average Tm. The degenerated primer has a minimal Tm of 51.3˚C and maximum Tm of 52.3˚C.

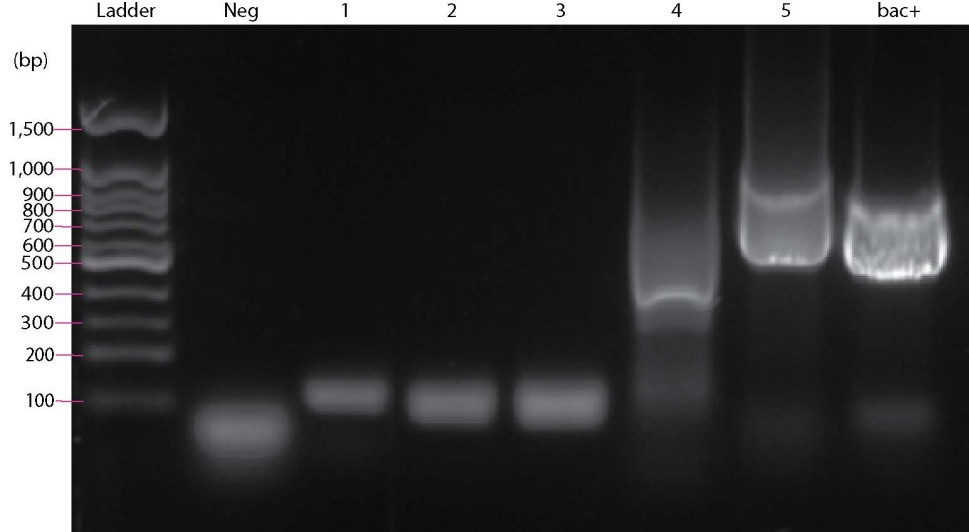

**Fig 5. Amplification of bacterial 16S, mammal and microsporidian rRNA in the _E. intestinalis_ ATCC 50506 spore sample obtained from HFF-1 cell culture.** _Ladder_: 100 bp ladder (FroggaBio, ON, Canada). _Neg_: negative control, no sample. _Well 1_: _E. intestinalis/E. hellem_ lsuF1-lsuR1 primers. _Well 2_: _E. intestinalis/E. hellem_ lsuF2-lsuR2 primers. _Well 3_: _E. intestinalis/E. hellem_ ssuF1-ssuR1 primers. _Well 4_: bacterial 357F-1100R primers. _Well 5_: mammal MF787- MR7535 primers. _Bac+_: _Staphylococcus salivarius_ as positive control for bacteria primers.

easily resuspended by inversion. Then, the clean microsporidian spores were pelleted by centrifugation (1,500 x _g_, 20 min, RT) and washed with 1 mL of PBS thoroughly (six times) to remove the DNase I from the sample and avoid microsporidian gDNA degradation during the downstream DNA extraction procedure. Spores were then resuspended in 1 mL of PBS and stored at 4°C. Microsporidian HMW DNA was extracted as resuspended in MBG water overnight as previously described [12,21].

### Validation of decrease in host DNA contamination in microsporidian gDNA sample

As aforementioned, the aim of this study was to develop optimized protocols to obtain microsporidian gDNA with minimal host DNA contaminant for whole genome sequencing purposes. To ensure that the gDNA sample obtained from purified microsporidian spores after DNase I treatment was enriched in microsporidian DNA, we qualitatively assessed the proportion of human:microsporidian DNA in the sample by amplifying the small/large rRNA subunit (SSU/LSU) gene(s) of the respective organisms by PCR [22].

PCR reactions were performed for a total of 35 cycles (initial denaturation: 95°C for 30 s; denaturation: 95°C for 30 s; annealing: 55°C for 30 s; elongation: 72°C for 2 min; final elongation: 72°C for 5 min) using universal mammalian (MF787-MR7535), universal microsporidian (ss530F-ss1047R) and _E. intestinalis_ specific (lsuF1-lsuR1, lsuF2-lsuR2, and ssuF1-ssuR1) primers (Table 2). We also used universal bacterial 16S primers (357F- 1100R) to identify possible contamination in the sample. PCR amplicons were verified by agarose gel electrophoresis (0.8%, TAE buffer 1X, 80 volts; Fig 6). The proportion of bacteria, human and microsporidian DNA in the sample was assessed by evaluating the relative intensities of the visible bands on agarose gel (Fig 6).

The PCR results showed that the microsporidian bands (Fig 6, wells 3–6) were the brightest on gel despite being much smaller in length (wells 3–5: ~100 bp; well 6: ~500 bp) than the mammalian amplicon (well 2: ~1,000 bp). These results indicated that microsporidian DNA was likely present in much higher proportions in the sample than contaminant DNA from the host and from bacteria, as calculated below.

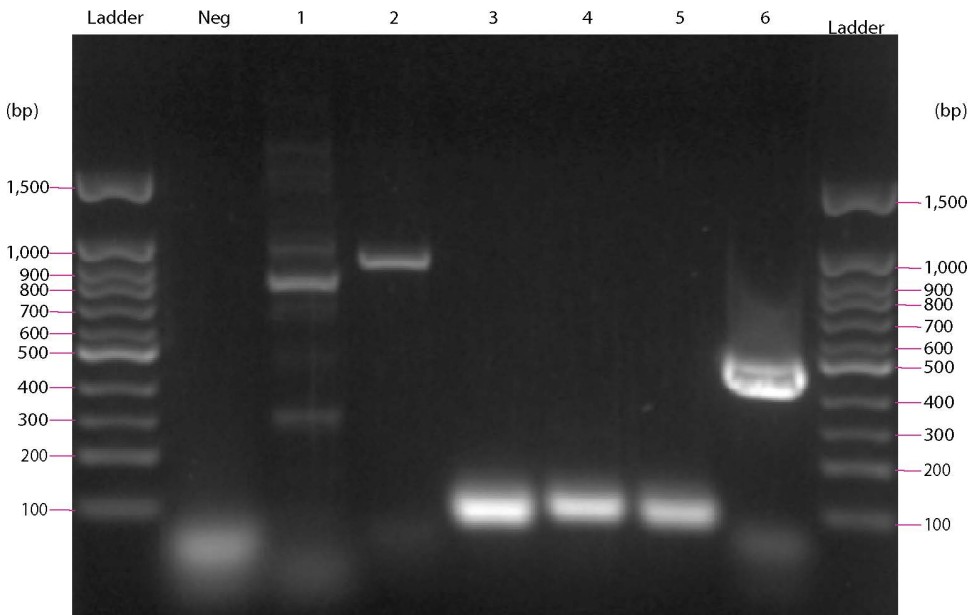

**Fig 6. PCR of gDNA sample extracted from *E. intestinalis* ATCC 50506 spores purified from infected HFF-1 cells post DNase I treatment.** *Ladder*: 100 bp ladder. *Neg*: negative control with all used primers. *Well 1*: universal bacterial primers for 16S gene (357F-1100R). *Well 2*: universal mammal primers for rRNA gene (MF787-MR7535). *Wells 3-5*: *E. intestinalis* specific primers for rRNA (lsuF1-lsuR1, lsuF2-lsuR2, and ssuF1-ssuR1). *Well 6*: universal microsporidian primers for small rRNA subunit (ss530F- ss1047R).

We estimated the relative proportion of DNA from microsporidia and the host by evaluating the number of rRNA gene copies per genome, the size difference of genomes, and the size difference of amplicons. First, the diploid human genome (from the HFF-1 cells) harbors about 400 copies of the rRNA genes whereas *Encephalitozoon* genomes encode 22 copies of the same rRNA genes per haploid copy [12]. This difference would cause the human DNA in the sample to be amplified roughly 20 times more than the microsporidian DNA at a 1:1 genome ratio (or ~10 times if the *Encephalitozoon* genomes are diploid). At 3 Gb, the human genome is also a thousand-fold bigger than that of *Encephalitozoon* species (~ 3 Mb). Because amplicons appear fainter than larger ones at a 1:1 ratio due to the smaller quantity of fluorescent dye being attached to the DNA, the tenfold difference in size between the amplicons from Fig 6 must also be factored in. Considering all the above, we estimated the percentage of microsporidian DNA in the sample as follows:

First, we estimated the proportion of microsporidia:human DNA in the sample (Equation 1) based on the results obtained from the PCR (Fig 6) using the assumption that *Encephalitozoon* spores genomes are diploid, as discussed by Selman *et al.* [23] and Khalaf *et al.* [24]. We qualitatively estimated that the gel band intensity (*i*) of the microsporidian DNA band (Fig 6 – well 3) was 10x higher than the human one (Fig 6 – well 2). We also considered the tenfold size difference(*s*) of the microsporidia (~ 100 bp) and human (~1,000 bp) amplicons. The relative rRNA copy number per genome copy was also considered in the calculations: *E. intestinalis* (44 rRNA copies per diploid genome; *EirRNA*) and human (400 rRNA copies per diploid genome; *HsrRNA*). The efficiency of the lsuF1-lsuR1 primers (*Eie*) was estimated by qPCR experiments as 35.7% and we hypothesized that the mammal primers had a near 100% efficiency (*Hse*) for calculation purposes. Thus:

$$i \; x \; s \; x \; \frac{HsrRNA}{EirRNA} \; x \; \frac{Hse}{Eie} = 10 \; x \; 10 \; x \; \frac{400}{44} \; x \; \frac{100}{35} = \mathbf{2,597.40}$$

(1)

 

Second, we considered the *E. intestinalis* (2.5 Mb) and human (3 Gb) genome sizes to determine the microsporidia:human DNA ratio in the sample. The previous calculation (Equation 1) showed that there were roughly 2,597 copies of the *E. intestinalis* genome per single copy of the human genome in the sample. Therefore:

$$2,597.4 \ x \ 2.5 \ \text{Mb} \ : \ 1 \ x \ 3\text{Gb}$$
$$6,492.5 \ : \ 3000$$
$$6.5 \ : \ 3 \tag{2}$$

Lastly, we calculated the percentage of *E. intestinalis* in the DNA sample by considering the 6.5:3 microsporidia:human DNA ratio (Equation 3). The total parts in the sample were 9.5 and *E. intestinalis* DNA corresponded to approximately 6.5 parts of it. Thus, from these approximations, *E. intestinalis* appeared to account for more than half of the gDNA sample.

$$\frac{6.5}{9.5} \ x \ 100 = 68.42\% \tag{3}$$

To verify this approximation computationally, we also independently evaluated the proportion of the bands in lanes 1, 2 and 6 (Fig 6) representing human, bacterial and microsporidian DNA by calculating the areas of the peaks present in each lane [25,26] with the ImageJ v1.53a [27] image processing tool. The ImageJ results (Fig 7) indicated that microsporidia, human and bacteria DNA composed approximately 77%, 15%, and 8% of the gDNA sample, respectively, suggesting that the quick and qualitative method used previously produced close estimates form the quantification method.

Since microsporidian DNA composed at least half of the sample composition, we sequenced the *E. intestinalis* genome using the three major high throughput sequencing technologies: Illumina, PacBio, and Oxford Nanopore. Table 3 summarizes the collected sequencing data and the proportions of microsporidia (reads that mapped) and potential contaminant

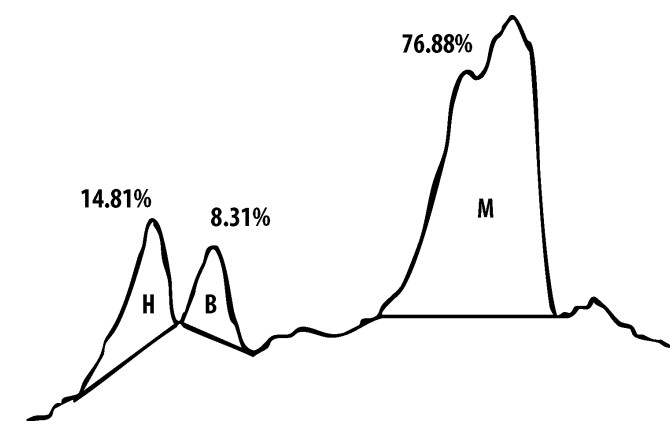

**Fig 7. Estimations of gel band proportions with ImageJ.** The proportions estimated for bands in electrophoresis for PCR of human **(H)**, bacterial **(B)** and microsporidian **(M)** DNA in the extracted gDNA sample.

**Table 3. Proportion of sequencing reads that mapped or did not map to the complete *E. intestinalis* ATCC 50506 genome.**

| Technology | Total read count | Mapped (%) | Not mapped (%) |
|---|---|---|---|
| Illumina (paired end) | 22,385,460 | 97.30 | 2.7 |
| PacBio | 8,951,401 | 89.91 | 10.09 |
| Oxford Nanopore | 1,464,740 | 83.30 | 16.7 |

DNA (reads that did not map) calculated through the mapping of sequencing reads against the reference *E. intestinalis* genome. Sequencing reads were mapped with get_SNPs.pl v2.0 from the SSRG pipeline (https://github.com/PombertLab/SSRG) using Minimap2 v2.22 as the read mapping tool [28]. The mapping results (Table 3) indicated that the protocols herein discussed did provide gDNA samples that were mostly composed of microsporidian DNA (83–97%), with minimal host-DNA contaminant. The Illumina dataset was the first one generated for the *E. intestinalis* ATCC 50506 genome, and it showed that at least 97.30% of the gDNA sample was from microsporidia.

As we anticipated, the calculations performed earlier underestimated the purity of the *E. intestinalis* gDNA sample due to the qualitative measures and assumptions we employed. This led us to use approximations about their respective efficiencies in the calculations that might have been off base (*e.g.,* the mammal primers are unlikely to be 100% efficient on the template DNA), but we used conservative estimates to prevent artificially inflating the amount of microsporidian DNA in our calculations. Following these results, we decided to move forward with sequencing this gDNA sample on the lab's Illumina MiniSeq instrument. This was done not solely to obtain short read data but also to experimentally validate the purity of our sample. The results obtained by Illumina sequencing confirmed that our protocol was working as intended, which led us to adopt it for all our sequencing experiments, including with the PacBio and Oxford Nanopore platforms. Altogether, the collected sequencing data was used to assemble the *E. intestinalis* 50506 genome from telomere-to-telomere (T2T), the first fully complete Microsporidia genome, followed by the *E. cuniculi* ATCC 50602*, E. hellem* ATCC 50604 and *E. hellem* ATCC 50451 complete T2T genomes (respective NCBI assembly accession numbers: ASM2439929v1, ASM2757158v1, ASM2439925v1 and ASM2921550v1) [12].

**Ploidy estimation in *E. intestinalis* ATCC 50506**

Following the near 100% representation of microsporidia in the sample sequenced with Illumina, we used this opportunity to estimate the ploidy status of *E. intestinalis* (strain ATCC 50506) based on the number of genome copies (*n*) per gDNA sample (Equation 4) [29]. The sample taken into consideration was of HMW gDNA (2,754 ng) obtained from 500 million pure spores. For this calculation, we used the following equation [29], wherein:

$$n = \frac{ng \times N_A}{L \times W \times 1 \times 10^9} \tag{4}$$

ng represents the total amount of DNA in the sample (2,754 ng). *NA* is the Avogadro's number (6.022 x $10^{23}$), which is defined by the number of molecules per mole of a substance. *L* is the estimated size of the *E. intestinalis* genome (2.5 Mbp). *W* is the average weight (650 Daltons) of a single DNA base pair. The bp unit was converted to ng by multiplying the genome size (in bp) to $1 \times 10^9$. Therefore:

$$n = \frac{2,754 \times 6.022 \times 10^{23}}{2,500,000 \times 650 \times 1 \times 10^9} = 1,020,590,031 \; copies$$

As observed, from 500 million *E. intestinalis* spores we obtained roughly 1 billion genome copies in the gDNA sample. If *E. intestinalis* was haploid, we would expect to obtain about 500 million genome copies from 500 million spores. Instead, our calculations suggest that the *E. intestinalis* 50506 is diploid, as suggested by Khalaf *et al.* [24].

**Conclusion**

The herein described protocols to culture different *Encephalitozoon* species *in vitro* were efficient to produce 1 billion spores after one month of culture. We obtained a total of 2.8 μg of gDNA from 500 million spores using bead-beating follow by the MasterPure Complete DNA Purification Kit (Biosearch Technologies, UK), which was sufficient DNA to perform sequencing with three different next generation sequencing (NGS) technologies. The protocols available in the literature

propose six months of microsporidia cell culture maintenance to obtain enough DNA for downstream procedures. Also, the high-throughput sequencing and resulting telomere-to-telomere confirmed that our gDNA sample was mostly composed of microsporidia. Altogether, our protocols were efficient in producing Microsporidian spores for DNA sequencing purposes, as evidenced by the resulting 4 telomere-to-telomere microsporidian genomes generated from it [12]. It is important to highlight that this protocol was optimized for gDNA extraction with minimal host DNA contaminant and might require further modifications if applied to other purposes.

## Supporting information

**S1 File. Step-by-step protocol, also available on protocols.io.**
(PDF)

## Author contributions

**Conceptualization:** Oscar X. Juárez, Jean-François Pombert, Karina Tuz.

**Data curation:** Anne Caroline Mascarenhas dos Santos, Jean-François Pombert.

**Formal analysis:** Anne Caroline Mascarenhas dos Santos, Jean-François Pombert, Karina Tuz.

**Funding acquisition:** Oscar X. Juárez, Jean-François Pombert, Karina Tuz.

**Investigation:** Anne Caroline Mascarenhas dos Santos, Pingdong Liang.

**Methodology:** Oscar X. Juárez, Karina Tuz.

**Project administration:** Jean-François Pombert, Karina Tuz.

**Resources:** Oscar X. Juárez, Jean-François Pombert, Karina Tuz.

**Supervision:** Karina Tuz.

**Validation:** Anne Caroline Mascarenhas dos Santos, Jean-François Pombert.

**Visualization:** Anne Caroline Mascarenhas dos Santos, Pingdong Liang, Jean-François Pombert.

**Writing – original draft:** Anne Caroline Mascarenhas dos Santos, Jean-François Pombert.

**Writing – review & editing:** Anne Caroline Mascarenhas dos Santos, Pingdong Liang, Oscar X. Juárez, Jean-François Pombert, Karina Tuz.

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
