## [Decision Letter · Decision Letter 0]

15 Dec 2025

Dear Dr. Tuz,

We look forward to receiving your revised manuscript.

Kind regards,

Olaf Kniemeyer

Academic Editor

PLOS One

Journal Requirements:

https://journals.plos.org/plosone/s/file?id=ba62/PLOSOne_formatting_sample_title_authors_affiliations.pdexf

2. We note you have not yet provided a protocols.io PDF version of your protocol and/or a protocols.io DOI. When you submit your revision, please provide a PDF version of your protocol as generated by protocols.io (the file will have the protocols.io logo in the upper right corner of the first page) as a Supporting Information file. The filename should be S1_file.pdf, and you should enter “S1 File” into the Description field. Any additional protocols should be numbered S2, S3, and so on. Please also follow the instructions for Supporting Information captions [https://journals.plos.org/plosone/s/supporting-information#loc-captions]. The title in the caption should read: “Step-by-step protocol, also available on protocols.io.”

Please assign your protocol a protocols.io DOI, if you have not already done so, and include the following line in the Materials and Methods section of your manuscript: “The protocol described in this peer-reviewed article is published on protocols.io (https://dx.doi.org/10.17504/protocols.io.[...]) and is included for printing purposes as S1 File.” You should also supply the DOI in the Protocols.io DOI field of the submission form when you submit your revision.

If you have not yet uploaded your protocol to protocols.io, you are invited to use the platform’s protocol entry service [https://www.protocols.io/we-enter-protocols] for doing so, at no charge. Through this service, the team at protocols.io will enter your protocol for you and format it in a way that takes advantage of the platform’s features. When submitting your protocol to the protocol entry service please include the customer code PLOS2022 in the Note field and indicate that your protocol is associated with a PLOS ONE Lab Protocol Submission. You should also include the title and manuscript number of your PLOS ONE submission.

“This work was supported by the National Institute of Allergy and Infectious Diseases of the National Institutes of Health [grant number R15AI128627] to J.-F.P. and [grant number 1R01AI151152-01A1] to O.J and K.T. The content is solely the responsibility of the authors and does not necessarily represent the official views of the National Institutes of Health”

“This work was supported by the National Institute of Allergy and Infectious Diseases of the National Institutes of Health [grant number R15AI128627] to J.-F.P. and [grant number 1R01AI151152-01A1] to O.J and K.T. The content is solely the responsibility of the authors and does not necessarily represent the official views of the National Institutes of Health”

Reviewers' comments:

Reviewer's Responses to Questions

**Comments to the Author**



Reviewer #1: Yes

Reviewer #2: Yes

2. Has the protocol been described in sufficient detail?

To answer this question, please click the link to protocols.io in the Materials and Methods section of the manuscript (if a link has been provided) or consult the step-by-step protocol in the Supporting Information files.

Reviewer #1: Partly

Reviewer #2: No

3. Does the protocol describe a validated method?

Reviewer #1: Yes

Reviewer #2: Yes

4. If the manuscript contains new data, have the authors made this data fully available?

Reviewer #1: Yes

Reviewer #2: N/A

**5. Is the article presented in an intelligible fashion and written in standard English?**

Reviewer #1: Yes

Reviewer #2: Yes

Reviewer #1: The protocol is interesting; however, it should be supplemented with additional information and some justifications. The authors suggest the use of human fibroblasts; however, what is the theoretical rationale for choosing this cell type for the microsporidium E. cuniculi? It is well known that this species exhibits tropism for canine or rabbit kidney cells. In addition, other microsporidia also develop well in these traditional cell lines. Why, then, was a non-standard cell line chosen instead of the commonly used ones? In my experience, determining the MOI (multiplicity of infection) is essential for the continuity and successful propagation of the cultures. However, the MOI was not identified in the manuscript. Another important point would be to explain why the spores present in the supernatant were not used. These spores are free of cellular content and are released after host cell lysis; therefore, they may yield higher-quality DNA with less contamination from host genomic material. It would be useful to highlight the spore-counting technique employed, as well as to indicate whether there were differences in spore production among the species—an aspect that we have also observed in our laboratory.

Reviewer #2: When I Googled "encephalitozoon culture protocol", two previously published protocols are in the top 5 hits: A Current Protocols paper from Han, et al (PMID: 30444582) and a protocols.io protocol from Antao, et al (DOI: 10.17504/protocols.io.3byl495wogo5/v1). The authors also cite two others in refs 8 and 10; I would agree that refs 8 and 10 aren't great step-by-step protocols with a lot of detail. But perhaps it would be fair to compare/contrast the Han, et al. and Antao, et al. protocols with the one from Mascarenhas dos Santos, et al? At first glance, this new protocol has much less detail regarding the Materials required. In my experience, ambiguity about reagents is 50% of the challenge of implementing a new method in the lab. Much of the rest of this method is fairly standard mammalian cell culture.

"Microsporidia" is frequently used as an adjective throughout the text. The authors also use "microsporidian", which I believe is the preferred adjective form. I would suggest using "microsporidian" throughout.

Ln 62-63: Is the "Expected Results" section missing? Or are these all sub-sections of the "Expected Results" section?

-Human foreskin fibroblast cell culture infected with Encephalitozoon spp.

-Purification of microsporidian spores

-DNase I treatment of microsporidia purified spores

-Validation of decrease in host DNA contamination in microsporidia gDNA sample

-Ploidy estimation in E. intestinalis ATCC 50506

If these are subsections, I was a little thrown because the sub-sections are mostly written in the past-tense, but I would have thought that expected results would be written in something more like the present or future tense? Just a small point and more of an editorial issue, but I found it confusing.

Ln 73-74: It would appear the glutamine is being added twice (first as part of PSQ, then again on its own). Correct?

To make this as straight-forward and accessible as possible to someone new to the culture of microsporidia, I suggest including sources and manufacturer and cat numbers for everything, and where possible, give concentrations etc in case the exact item is no longer available. For example, the authors do a nice job of this for PSQ (lns 71-74). However, this info is missing in many other places. For example, DMEM comes in so many formulations with or without pyruvate, glutamine, etc; it would be helpful to describe the key features (e.g., plus x mM sodium pyruvate), and the supplier/cat#. Same goes for the Petri dishes, PBS, gelatin, trypsin/EDTA (how much EDTA?), etc. Perhaps there should be a Materials table, with absolutely everything required, plus supplier/cat#?

Ln 140: Perhaps it would be more appropriate to call this section "Isolation of microsporidian spores"? I don't think I would call this a purification.

Pgs 13-14: This section seems a bit weak. This seems like something that should be addressed by qPCR. If the primer sets for human vs Encephalitozoon amplicons differ in efficiency, couldn't the ratio of the two products obtained be skewed substantially one way or the other? I feel it isn't rigorous to try to assess the ration of the two genomes in this way.

Fig 4: Can a higher quality image be obtained? This appears to be out of focus and not particularly high res. It is unclear what features allow spores (arrows) to be distinguished from the other bodies on the slide.

**Do you want your identity to be public for this peer review?** For information about this choice, including consent withdrawal, please see our Privacy Policy

Reviewer #1: **Yes:** Maria Anete Lallo

Reviewer #2: No

---

## [Author Response · Author response to Decision Letter 1]

2 Feb 2026

Response to the editor

This revised version of the manuscript incorporates and addresses all reviewers’ feedback, resulting in a strengthened and more clearly articulated study. We have added a section on how our protocol differs from previously published articles and have provided a rationale for the selection of human fibroblasts as the host model. All reviewer comments have been fully addressed both within the revised manuscript and in a separate document titled Response to Reviewers, as instructed.

Out manuscript has also been formatted according to the instructions of the journal. We have assigned our protocol a protocols.io DOI and added this information to the manuscript as per Jounal’s instructions.

We have updated our Funding Statement to:

“This work was supported by the National Institute of Allergy and Infectious Diseases of the National Institutes of Health [grant number R15AI128627] to J.-F.P. and [grant number 1R01AI151152-01A1] to O.J and K.T. The content is solely the responsibility of the authors and does not necessarily represent the official views of the National Institutes of Health. The founders had no role in study design, data collection and analysis, decision to publish, or preparation of the manuscript. There was no additional funding received for this study.”

Response to reviewers

Reviewer #1: The protocol is interesting; however, it should be supplemented with additional information and some justifications. The authors suggest the use of human fibroblasts; however, what is the theoretical rationale for choosing this cell type for the microsporidium E. cuniculi? It is well known that this species exhibits tropism for canine or rabbit kidney cells. In addition, other microsporidia also develop well in these traditional cell lines. Why, then, was a non-standard cell line chosen instead of the commonly used ones? In my experience, determining the MOI (multiplicity of infection) is essential for the continuity and successful propagation of the cultures. However, the MOI was not identified in the manuscript. Another important point would be to explain why the spores present in the supernatant were not used. These spores are free of cellular content and are released after host cell lysis; therefore, they may yield higher-quality DNA with less contamination from host genomic material. It would be useful to highlight the spore-counting technique employed, as well as to indicate whether there were differences in spore production among the species—an aspect that we have also observed in our laboratory.

Answer:

We thank the reviewer for their feedback. We have incorporated their suggestions to the manuscript.

• Murine and human fibroblasts cells, lung and foreskin, are readily infected with Encephalitozoon spp. This rationale has been added to the manuscript. Lines 67-68.

• The MOI was in the range of 0.5 to 20, this information has been incorporated to the manuscript, line 99.

• The number of spores harvested from any Encephalitozoon spp. described in this work consistently provided enough DNA material for sequencing analysis after infection for one month. The quantity of DNA we obtained from infections was sufficient to sequence their full-length genomes with multiple sequencing technologies. The number of spores in the supernatant were less abundant than those obtained from the host cell lysis. Additionally, the spores from the supernatant were meant to be used as stock for later propagation in a follow up study as the spores were fully matured and ready for germination/infection, a certainty not possible from the Encephalitozoon spp. obtained from infected cell lines, as they are present in multiple life stages inside the host cells and not only in the form of mature spores suitable for re-infection of cell lines.

• Spores were diluted and counted using a hemocytometer and a 1:100 dilution ratio as described in the manuscript, lines 166-167.

• No differences in spore production were observed among the three Encephalitozoon species evaluated. Spore production in HFF-1 cells was allowed for a month which yielded enough spores for DNA isolation for downstream genomic analyses. Lines 140-141.

Reviewer #2: When I Googled "encephalitozoon culture protocol", two previously published protocols are in the top 5 hits: A Current Protocols paper from Han, et al (PMID: 30444582) and a protocols.io protocol from Antao, et al (DOI: 10.17504/protocols.io.3byl495wogo5/v1). The authors also cite two others in refs 8 and 10; I would agree that refs 8 and 10 aren't great step-by-step protocols with a lot of detail. But perhaps it would be fair to compare/contrast the Han, et al. and Antao, et al. protocols with the one from Mascarenhas dos Santos, et al? At first glance, this new protocol has much less detail regarding the Materials required. In my experience, ambiguity about reagents is 50% of the challenge of implementing a new method in the lab. Much of the rest of this method is fairly standard mammalian cell culture.

Answer:

We thank the reviewer for their constructive feedback.

Han et al. and Antao et al. are the most recent and detailed protocols for the culturing and propagation of Encephalitozoon spp. in vitro. Although their work greatly increased transparency, availability and reproducibility of the protocols to cultivate Microsporidia species in the laboratory, we still found ourselves in need of more specific measurements for both cultivation and spore/DNA purification yields.

Han et al. (PMID: 30444582) did an excellent work at describing the process to culture Encephalitozoon spp. in vitro in detail. Although their work was thorough, it did not address the levels of microsporidia vs host DNA in the samples, or available treatments to increase the yield of microsporidian DNA post culturing. As observed in Han and other published protocols, the exact length of culturing and quantity of spores produced from their methods is not reported, which makes it infeasible for us to make comparisons on the efficiency of protocols when those measures are not available to the community. However, in our protocol we ensured that the estimation of produced spores was made available (spore yield after 1 month of cell culturing in 10 petri dishes) so that we start building a frame of reference on the efficiency and yields of Microsporidia cultivation in vitro.

Antao et al. (DOI: 10.17504/protocols.io.3byl495wogo5/v1) measured the quantity of E. intestinalis DNA in their samples post-purification using a standard percoll gradient protocol, which showed that in best-case scenario the microsporidian gDNA in the sample could achieve a purity of 80%, in contrast to the protocols we described using the DNase treatment, our sequencing data revealed that a minimum purity of 80% for microsporidian DNA when being conservative and using error-prone Oxford Nanopore Sequencing data for the estimations and reached over 90% when estimated with the accurate Illumina data. This demonstrates that our protocol can deliver high-purity microsporidia DNA, but it is important to note that comparisons are not direct since Antao et al. estimated purity of DNA using only qPCR instead of genome sequencing.

We have included a brief protocol discussion of the Han et al. and Antao et al. protocols in the manuscript, lines 48-51.

The specific materials used have been incorporated now into the appropriate manuscript sections and summarized in the new Table 1.

Reviewer #2:

"Microsporidia" is frequently used as an adjective throughout the text. The authors also use "microsporidian", which I believe is the preferred adjective form. I would suggest using "microsporidian" throughout.

Answer:

The use of Microsporidia as an adjective has been corrected throughout the manuscript text.

Reviewer #2:

Ln 62-63: Is the "Expected Results" section missing? Or are these all sub-sections of the "Expected Results" section?

-Human foreskin fibroblast cell culture infected with Encephalitozoon spp.

-Purification of microsporidian spores

-DNase I treatment of microsporidia purified spores

-Validation of decrease in host DNA contamination in microsporidia gDNA sample

-Ploidy estimation in E. intestinalis ATCC 50506

If these are subsections, I was a little thrown because the sub-sections are mostly written in the past-tense, but I would have thought that expected results would be written in something more like the present or future tense? Just a small point and more of an editorial issue, but I found it confusing.

Answer:

For clarity, the “Expected Results” head has been removed. Each section stands on its own and brings overall clarity.

Reviewer #2:

Ln 73-74: It would appear the glutamine is being added twice (first as part of PSQ, then again on its own). Correct?

Answer:

Yes, glutamine was added as part of the PSQ supplement and then individually.

Reviewer #2:

To make this as straight-forward and accessible as possible to someone new to the culture of microsporidia, I suggest including sources and manufacturer and cat numbers for everything, and where possible, give concentrations etc in case the exact item is no longer available. For example, the authors do a nice job of this for PSQ (lns 71-74). However, this info is missing in many other places. For example, DMEM comes in so many formulations with or without pyruvate, glutamine, etc; it would be helpful to describe the key features (e.g., plus x mM sodium pyruvate), and the supplier/cat#. Same goes for the Petri dishes, PBS, gelatin, trypsin/EDTA (how much EDTA?), etc. Perhaps there should be a Materials table, with absolutely everything required, plus supplier/cat#?

Answer:

We thank the reviewer for the suggestion. Table 1 lists the materials used for HFF-1 cultivation and microsporidian spore isolation (Line 160).

Reviewer #2:

Ln 140: Perhaps it would be more appropriate to call this section "Isolation of microsporidian spores"? I don't think I would call this a purification.

Answer:

The section name has been changed to “Isolation of microsporidian spores” (Line 146)

Reviewer #2:

Pgs 13-14: This section seems a bit weak. This seems like something that should be addressed by qPCR. If the primer sets for human vs Encephalitozoon amplicons differ in efficiency, couldn't the ratio of the two products obtained be skewed substantially one way or the other? I feel it isn't rigorous to try to assess the ration of the two genomes in this way.

Answer:

We thank the reviewer for the comment. We do agree with the reviewer that estimating the ratios of microsporidia vs host DNA in the samples through PCR is not an ideal approach, and here our only intent was to observe an overall trend of sample composition to confidently deploy DNA sequencing for an unbiased estimation of the sample purity. Contrary to PCR and qPCR, genome sequencing allowed us to look at all DNA in our samples and capture not only accurate proportions of microsporidia vs host DNA by providing access to complete genome sequences but also gave us the capability to inspect for contaminants, for a comprehensive sample composition/purity assessment. The sample composition and estimation of microsporidia DNA in the samples was reported in table 3 for all three sequencing technologies deployed in the validation of the protocols.

Reviewer #2:

Fig 4: Can a higher quality image be obtained? This appears to be out of focus and not particularly high res. It is unclear what features allow spores (arrows) to be distinguished from the other bodies on the slide.

Answer:

We are submitting a new figure 4 with increased quality (600 dpi instead of 300 dpi) where all spores have been pointed at with arrows.

---

## [Decision Letter · Decision Letter 1]

22 Feb 2026

In vitro culture of human-infecting Encephalitozoon spp.for genome sequencing with minimal host contaminant

PONE-D-25-57968R1

Dear Dr. Tuz,

We’re pleased to inform you that your manuscript has been judged scientifically suitable for publication and will be formally accepted for publication once it meets all outstanding technical requirements.

Kind regards,

Olaf Kniemeyer

Academic Editor

PLOS One

Additional Editor Comments (optional):

Reviewers' comments:

Reviewer's Responses to Questions

**Comments to the Author**



Reviewer #1: Yes

Reviewer #2: Yes

2. Has the protocol been described in sufficient detail?

To answer this question, please click the link to protocols.io in the Materials and Methods section of the manuscript (if a link has been provided) or consult the step-by-step protocol in the Supporting Information files.

Reviewer #1: Yes

Reviewer #2: Yes

3. Does the protocol describe a validated method?

Reviewer #1: Yes

Reviewer #2: Yes

4. If the manuscript contains new data, have the authors made this data fully available?

Reviewer #1: Yes

Reviewer #2: Yes

**5. Is the article presented in an intelligible fashion and written in standard English?**

Reviewer #1: Yes

Reviewer #2: Yes

Reviewer #1: The recommended corrections and clarifications were carried out by the authors, and the adjustments improved the understanding of the protocol to be published.

Reviewer #2: The authors have addressed my major concerns.

**Do you want your identity to be public for this peer review?** For information about this choice, including consent withdrawal, please see our Privacy Policy

Reviewer #1: **Yes:** Profa. Dra. Maria Anete Lallo

Reviewer #2: No

---

## [Editor Report · Acceptance letter]

PONE-D-25-57968R1

PLOS One

Dear Dr. Tuz,

I'm pleased to inform you that your manuscript has been deemed suitable for publication in PLOS One. Congratulations! Your manuscript is now being handed over to our production team.

Kind regards,

on behalf of

Dr. Olaf Kniemeyer

Academic Editor

PLOS One